# An Observation of the Vitamin D Status in Highly Trained Adolescent Swimmers during the UK Autumn and Winter Months

Josh W. Newbury [1], Meghan A. Brown [2], Matthew Cole [1], Adam L. Kelly [1] and Lewis A. Gough [1,*]

[1] Research Centre for Life and Sport Science (CLaSS), School of Health Sciences, Birmingham City University, Birmingham B15 3TN, UK; josh.newbury@bcu.ac.uk (J.W.N.); matthew.cole@bcu.ac.uk (M.C.); adam.kelly@bcu.ac.uk (A.L.K.)

[2] Carnegie School of Sport, Leeds Beckett University, Leeds LS6 3QQ, UK; meghan.brown@leedsbeckett.ac.uk

* Correspondence: lewis.gough@bcu.ac.uk

**Abstract:** The purpose of this research was two-fold: (a) to observe whether highly trained adolescent swimmers abide to vitamin D supplement recommendations; and (b) to monitor changes in circulating 25-hydroxyvitamin D (25(OH)D) that occur between the autumn and winter months. Twenty swimmers (age: $17 \pm 2$ years) from a UK high-performance swimming club volunteered to complete two blood spot cards to determine their 25(OH)D concentration: the first in an autumn training phase (October) and the second during winter training (January). All swimmers were advised to consume vitamin $D_3$ supplements across the assessment period; however, only 50% of swimmers adhered to this recommendation. Resultantly, a winter decline in 25(OH)D was observed in non-supplementing swimmers ($79.6 \pm 25.2$ to $52.6 \pm 15.1$ nmol·L$^{-1}$, $p = 0.005$), with swimmers either displaying an 'insufficient' (60%) or 'deficient' (40%) vitamin D status. In comparison, a greater maintenance of 25(OH)D occurred in supplementing swimmers ($92.0 \pm 25.5$ to $97.2 \pm 38.3$ nmol·L$^{-1}$, $p = 0.544$), although variable outcomes occurred at the individual level (four increased, three maintained, three declined). These findings highlight the possible risks of vitamin D insufficiency during the winter for swimmers in the UK, possibly requiring standardised supplement practices. Moreover, alternative educational strategies may be required for swimmers to transfer knowledge to practice in order to improve supplement adherence in future.

**Keywords:** vitamin D; supplements; sport nutrition; swimming; adolescent athletes

## 1. Introduction

Vitamin D is a fat-soluble vitamin with an involvement in numerous physiological processes, including bone health, immunity, cardiac function, and skeletal muscle remodelling [1], all of which could support the long-term health and performance of swimmers. However, consuming an adequate vitamin D intake through diet alone can be challenging, considering that low amounts of bioavailable vitamin D (i.e., ergocalciferol, cholecalciferol) are naturally found in foods and beverages [2]. In contrast, large quantities of vitamin D can be naturally produced following direct sun exposure [3]. This occurs as the sun's ultraviolet-B (UVB) radiation interacts with 7-dehydrocholesterol in the skin to catalyse the formation of cholecalciferol (vitamin $D_3$), which is later converted into 25-hydroxyvitamin D (25(OH)D) in the liver [2]. Yet, as this process is dependent on achieving UVB exposure, it can become limited by two key factors: (a) living and training in countries of northern latitudes ($\geq 40°$ N), whereby an 80–100% decrease in UVB availability occurs in the autumn and winter months [4,5]; and (b) spending large quantities of time indoors, either through training, school, and/or employment [6]. This produces a problem for highly trained adolescent swimmers that reside in the UK (latitude: 51–55° N), who often fail to consume the UK reference nutrient intake for vitamin D (<10 µg·day$^{-1}$), and spend ~15–20 h·week$^{-1}$ training indoors [7]. Thus, for this population, the use of vitamin $D_3$ supplements is warranted.

Swimming as a sport is associated with a large seasonal decline in circulating 25(OH)D concentration, although this can be offset with vitamin $D_3$ supplements. Indeed, supplementation with 4000 $IU \cdot day^{-1}$ vitamin $D_3$ maintained the 25(OH)D concentrations of collegiate swimmers over an autumn and winter training period (August to March: $+2.5$ $nmol \cdot L^{-1}$), compared to a 31% decline when swimmers consumed a placebo supplement ($-50$ $nmol \cdot L^{-1}$) [8]. Similarly, an intake of 5000 $IU \cdot day^{-1}$ vitamin $D_3$ was also found to increase the 25(OH)D of collegiate swimmers across an autumn training period (August to November: $+9$ $nmol \cdot L^{-1}$), with a large decline observed in swimmers who ingested a placebo supplement ($-40$ $nmol \cdot L^{-1}$) [9]. However, the locations within these studies (USA: 37–38° N) meant that most swimmers started the winter period with a high starting 25(OH)D concentration ($>120$ $nmol \cdot L^{-1}$), enabling a 'sufficient' amount of circulating vitamin D ($\geq 75$ $nmol \cdot L^{-1}$) to be maintained despite experiencing large declines [10]. In contrast, 66% of adolescent swimmers from Israel (31° N) were found to have 'insufficient' 25(OH)D ($<50$ $nmol \cdot L^{-1}$) during the autumn months (October: $62 \pm 12$ $nmol \cdot L^{-1}$), which declined to levels close to 'deficiency' by the winter without the use of vitamin $D_3$ supplements (January: $51 \pm 11$ $nmol \cdot L^{-1}$) [11]. Moreover, even with the use of 2000 $IU \cdot day^{-1}$ vitamin $D_3$ supplements, only 48% of a supplementing sub-group achieved a 'sufficient' vitamin D status, highlighting the possible risks of vitamin D insufficiency in adolescent swimmers [11]. This combined evidence shows that large seasonal declines in 25(OH)D occur in swimmers regardless of location, prompting the supplementation of 2000–5000 $IU \cdot day^{-1}$ vitamin $D_3$ from August to March.

The importance of vitamin D and the risks of deficiency have become well acknowledged in recent years, with a large proportion of athletes (72–97%) now recognising the possible health and performance benefits of supplementation [12–14]. Despite greater educational provisions, however, it is currently unclear whether swimmers now adhere to supplement recommendations, or if widespread seasonal declines in serum 25(OH)D still exist. For example, only 56% of adolescent swimmers in Denmark declared the use of vitamin $D_3$ supplements during the winter months [15], even though high risks of deficiency were present in this population (e.g., latitude: 55° N, indoor training volume: 30 $h \cdot week^{-1}$ [6]). Furthermore, those that did supplement used a wide variety of vitamin $D_3$ dosages (mean: $2600 \pm 1960$ $IU \cdot day^{-1}$), resulting in both supplement users ($57 \pm 21$ $nmol \cdot L^{-1}$) and non-users ($39 \pm 13$ $nmol \cdot L^{-1}$) displaying 'insufficient' and 'deficient' 25(OH)D concentrations, respectively [1]. In addition, results from a recent study suggested that only 73% of national-level swimmers, and 38% of age-group (aged 13–17 years) swimmers in the UK currently utilise vitamin $D_3$ supplements, even after receiving education and individual nutrition support [16], though it was unclear how this lack of supplementation affected circulating 25(OH)D concentrations. Hence, the aim of this study was to assess the serum 25(OH)D concentrations in a cohort of UK-based, highly trained adolescent swimmers at two in-season time points: in the autumn (October), and during the winter (January).

## 2. Materials and Methods

### 2.1. Participants

Twenty adolescent swimmers from a high-performance swimming club in the UK volunteered for this study (Table 1), which an *a priori* power calculation determined to be an appropriate sample size for identifying moderate effect sizes (0.50) in within–between interactions, repeated-measures analysis of variance (ANOVA) tests (two groups, two measurements) with a power $>80\%$ (input parameters: $\alpha = 0.05$, $\beta = 0.80$, correspondence = 0.3; G*Power, v.3.1.9.4, Universität Düsseldorf, Germany). All swimmers were competitive at the national level in the UK within their respective age categories and were all therefore classified as 'highly trained' [17]. Moreover, seven swimmers from this cohort had recently represented their nations at junior international competitions. At the time of the study, swimmers were completing between 5–9 pool and 2–5 gym-based training sessions$\cdot week^{-1}$ at their training facility in the West Midlands, UK (latitude: 52° N), where they remained throughout the observation period.

**Table 1.** Characteristics of the study participants.

| Swimmers | Age (years) | Height (m) | Body Mass (kg) | WA Points * |
|---|---|---|---|---|
| **Male (*n* = 8)** | 18 ± 2 | 1.80 ± 0.04 | 72.6 ± 8.3 | 705 ± 83 |
| **Female (*n* = 12)** | 16 ± 2 | 1.70 ± 0.09 | 62.1 ± 6.9 | 690 ± 55 |
| **Combined (*n* = 20)** | 17 ± 2 | 1.74 ± 0.09 | 66.3 ± 9.0 | 696 ± 66 |

* Mean World Aquatic points scored for the swimmers' best event in a long-course (50 m) swimming competition [18]. Mean ± standard deviation.

*2.2. Experimental Procedures*

Based on the adolescent cohort, dried blood spot cards were used to determine serum 25(OH)D concentrations due to their close agreement ($r = 0.74$–$0.97$), yet less invasive procedures compared to venipuncture methods [19–22]. The collection process required four fingertip capillary blood drops (~50–75 μL), which were spotted onto a filter paper at four equally spaced targets to allow for radial dispersion. All blood collection occurred as the swimmers arrived at their normal swimming training; therefore, cards were sealed and allowed to dry at room temperature for between 2 and 3 h until the end of the training session. Samples were then posted to an independent laboratory (Sandwell & West Birmingham Hospitals NHS Trust, Birmingham, UK), where they were analysed using liquid chromatography–tandem mass spectrometry (LC-MS/MS) within 7–14 days. This process was completed twice per swimmer to assess possible seasonal changes in vitamin D status. These were completed at mid-season training phases in the autumn (first week of October) and winter (first week of January).

For ethical and performance reasons, all swimmers and care givers were given basic information regarding the importance of vitamin D prior to the winter months. This was delivered by the lead researcher to all swimmers as part of a classroom-based presentation (~20 min), whereas care givers were sent the presentation slides electronically via a group instant messaging application (WhatsApp, Menlo Park, CA, USA). Within the slides was the recommendation to supplement with 2000–5000 IU·day$^{-1}$ vitamin $D_3$ from October until March, which was suggested based on previous research in swimmers [8,9,11]. Following the winter (January) measurement of 25(OH)D, swimmers were asked whether they had taken any vitamin $D_3$ supplements over the observation period. Ten (50%) swimmers declared the use of vitamin $D_3$ supplements, albeit at varying doses (2 × 400 IU·day$^{-1}$, 1 × 1000 IU·day$^{-1}$, 1 × 2000 IU·day$^{-1}$, 5 × 2500 IU·day$^{-1}$, 1 × 4000 IU·day$^{-1}$). Conveniently, a sub-group analysis of supplementing (VITD) versus non-supplementing (NONE) swimmers was included in the results.

For the purpose of this study, the following vitamin D language is used in accordance with the Endocrine Society guidelines [23], as per previous reviews in athletic cohorts [24–26]: serum 25(OH)D $\geq 75$ nmol·L$^{-1}$ = 'sufficient'; 50–74 nmol·L$^{-1}$ = 'insufficient'; 25–49 nmol·L$^{-1}$ = 'deficient'; and $< 25$ nmol·L$^{-1}$ = 'severely deficient'.

*2.3. Statistical Analysis*

All data were normally distributed (Shapiro–Wilk) and checked for sphericity (Mauchly) prior to parametric testing. A 2 × 2 repeated-measures ANOVA was used to determine main effects of time for the whole group of swimmers (October vs. January), as well as group-level interactions between the VITD (*n* = 10) and NONE (*n* = 10) sub-groups. For statistically significant results, the Bonferroni correction was applied to identify time (October, January)–treatment (VITD, NONE) interactions between the sub-groups. Statistical significance was accepted at $p < 0.05$. If sphericity was violated during the ANOVA, then the appropriate Hyun–Feldt (epsilon value >0.75) or Greenhouse–Geiser (epsilon value <0.75) corrections were applied. Effect sizes for ANOVA tests were reported as partial eta squared ($P\eta^2$) values, which were interpreted as 'small' (0.01–0.05), 'moderate' (0.06–0.13), and 'large' ($\geq 0.14$) [27]. Effect sizes for pairwise comparisons were also calculated using the Hedge's *g* bias correction, which accounted for the bias in Cohen's *d* with small (*n* $\leq$ 20) study samples [28]. These effect sizes (*g*) were interpreted as 'trivial' ($\leq 0.19$), 'small' (0.20–0.49), 'moderate' (0.50–0.79), and

'large' ($\geq$0.80) [27]. A smallest worthwhile change (SWC) of $\pm$4.3 nmol$\cdot$L$^{-1}$ was calculated by multiplying the standard deviation of the initial October data set by 0.2 [29]. All data are reported as mean $\pm$ standard deviation. All statistical analyses were carried out using Statistical Package for Social Sciences (v.28, IBM, New York, NY, USA) and $p < 0.05$ was accepted for significance. Data are reported as mean $\pm$ standard deviation.

### 3. Results

At the group mean level, 25(OH)D concentrations reduced by 13% from October (86 $\pm$ 26 nmol$\cdot$L$^{-1}$) to January (75 $\pm$ 36 nmol$\cdot$L$^{-1}$), but this decline did not reach statistical significance ($p = 0.082$, $P\eta^2 = 0.16$). However, large individual variances in serum 25(OH)D were observed at both the October (range: 46–124 nmol$\cdot$L$^{-1}$) and January time points (range: 31–172 nmol$\cdot$L$^{-1}$), which masked the changes taking place at the individual level. Overall, 16 swimmers (80%) experienced changes in serum 25(OH)D that exceeded the SWC, including 12 swimmers (60%) who experienced declines (range: $-6$ to $-63$ nmol$\cdot$L$^{-1}$) and four swimmers (20%) who experienced increases (range: +10 to +52 nmol$\cdot$L$^{-1}$). Based on these individual changes, there was an increase in the number of swimmers who developed an 'insufficient' vitamin D status from October ($n = 8$) to January ($n = 14$), including a greater number of swimmers who could be classified as 'deficient' (two in October, five in January).

*Sub-Group Analysis*

The highly variable changes in vitamin D status observed at the group level were due to differences in supplement intakes ($p = 0.014$, $P\eta^2 = 0.29$; Figure 1). While there was little difference between sub-groups in October (VITD: 92 $\pm$ 25 nmol$\cdot$L$^{-1}$ vs. NONE: 80 $\pm$ 25 nmol$\cdot$L$^{-1}$, $p = 0.228$, $g = 0.47$), the NONE sub-group experienced a significant decline in 25(OH)D by January (mean change: $-27 \pm 20$ nmol$\cdot$L$^{-1}$, $p = 0.005$, $g = 1.24$). In contrast, the VITD sub-group had better maintenance of their 25(OH)D concentration (mean change: +5 $\pm$ 31 nmol$\cdot$L$^{-1}$, $p = 0.544$, $g = 0.15$). This resulted in a large difference between the two sub-groups at the January time point (VITD: 97 $\pm$ 38 nmol$\cdot$L$^{-1}$ vs. NONE: 53 $\pm$ 15 nmol$\cdot$L$^{-1}$, $p = 0.005$, $g = 1.47$). Nine swimmers in the NONE sub-group experienced declines in serum 25(OH)D above the SWC, resulting in all NONE swimmers having 'insufficient' or 'deficient' vitamin D in January (Table 2). In contrast, variable 25(OH)D changes occurred in the VITD group (four increased, three maintained, three declined), likely occurring because of the variable supplement strategies.

**Table 2.** Individual changes in serum 25(OH)D concentration from October to January in swimmers who reported (VITD) or who did not report (NONE) consuming vitamin D$_3$ supplements.

| | NONE | | | VITD | |
|---|---|---|---|---|---|
| **Swimmer** | **Vitamin D$_3$ Dose (IU$\cdot$Day$^{-1}$)** | **$\Delta$ from October (nmol$\cdot$L$^{-1}$)** | **Swimmer** | **Vitamin D$_3$ Dose (IU$\cdot$Day$^{-1}$)** | **$\Delta$ from October (nmol$\cdot$L$^{-1}$)** |
| **1** | 0 | $-26.3$ | **11** | 400 | $-5.4$ |
| **2** | 0 | $-53.2$ | **12** | 2500 | +25.0 |
| **3** | 0 | $-53.0$ | **13** | 4000 | +41.9 |
| **4** | 0 | $-54.6$ | **14** | 2000 | +9.5 |
| **5** | 0 | $-23.8$ | **15** | 2500 | $-3.7$ |
| **6** | 0 | $-5.5$ | **16** | 1000 | $-62.9$ |
| **7** | 0 | $-19.9$ | **17** | 2500 | +3.1 |
| **8** | 0 | +1.0 | **18** | 2500 | +52.0 |
| **9** | 0 | $-14.3$ | **19** | 2500 | $-6.3$ |
| **10** | 0 | $-19.9$ | **20** | 400 | $-1.4$ |

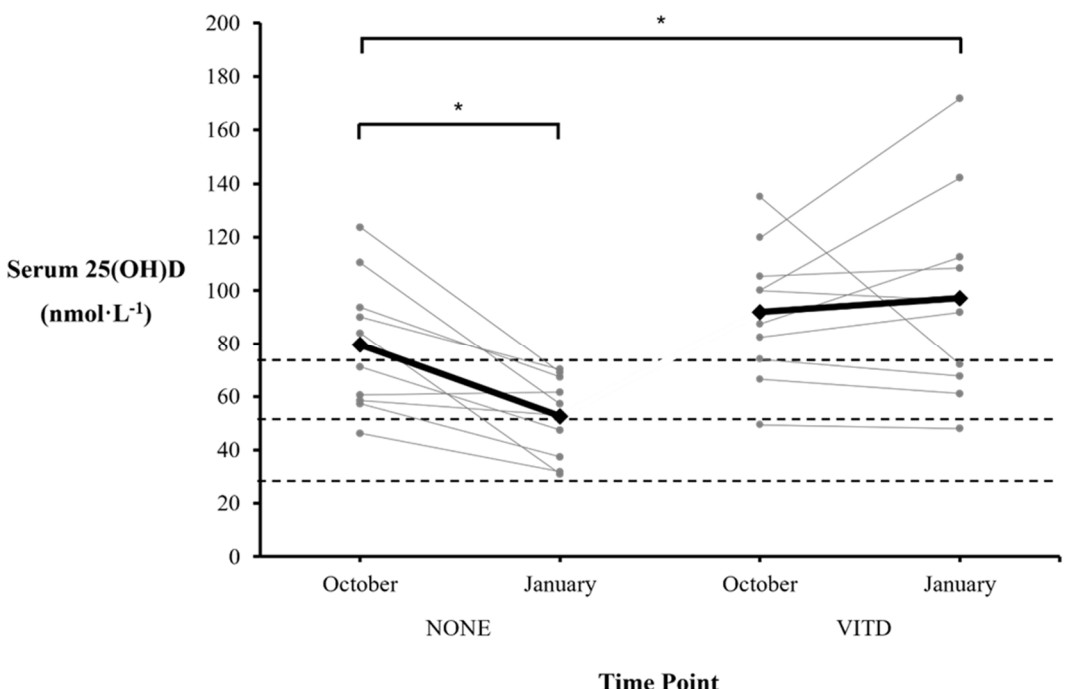

**Figure 1.** Individual changes in serum 25(OH)D concentration from October to January in swimmers who reported using vitamin $D_3$ supplements (VITD) or no supplements (NONE). Black line represents sub-group means. * = between or within group differences ($p < 0.05$). Dotted lines represent vitamin D thresholds: $\leq 25$ nmol·$L^{-1}$ = 'severely deficient'; 25–49 nmol·$L^{-1}$ = 'deficient'; 50–74 nmol·$L^{-1}$ = 'insufficient'; $\geq 75$ nmol·$L^{-1}$ = 'sufficient' [23].

## 4. Discussion

This was the first study to observe a seasonal change in the vitamin D status of highly trained adolescent swimmers in the UK. A concerning finding was that only 60% of swimmers displayed a sufficient vitamin D status in October, which was approximately one month following a summer break from training. Moreover, all swimmers were advised to supplement with 2000–5000 IU·day$^{-1}$ across the autumn and winter months in accordance with previous research [8,9,11], although only 50% of the swimmers adhered to this recommendation. This resulted in highly variable changes in serum 25(OH)D occurring across the observation window, with swimmers either increasing ($n = 4$), maintaining ($n = 4$), or declining ($n = 12$) in vitamin D status. Importantly, the majority of swimmers who experienced 25(OH)D declines were identified in the sub-group who reported using no vitamin $D_3$ supplements ($n = 9$), resulting in all swimmers in this sub-group having an 'insufficient' (60%) or 'deficient' (40%) vitamin D status at a mid-season winter time point. These results suggest that almost swimmers in the UK may benefit from using vitamin $D_3$ supplements in the autumn and winter months, although further research is needed to identify methods to increase adherence to the recommendations.

From a group mean perspective, highly trained adolescent swimmers maintained a 'sufficient' vitamin D status at both the October and January time points; however, this analysis masked that 80% of the cohort experienced changes in 25(OH)D that exceeded the SWC ($\pm 4.3$ nmol·$L^{-1}$). This failure to detect whole-group changes in vitamin D status occurred since 50% of cohort avoided using vitamin $D_3$ supplements in the autumn and winter months, whereas the other 50% used vitamin $D_3$ supplements of varying doses (400–4000 IU·day$^{-1}$). This was in accordance with research by Geiker et al. [15], who also showed that highly trained adolescent swimmers do not adhere to supplement recommendations, which in turn, resulted in a large proportion of swimmers developing 'insufficient' and 'deficient' 25(OH)D concentrations across the winter months. Indeed, based on whole group data, 70% ($n = 14$) of the current UK-based cohort were found to have 'insufficient'

vitamin D in January, supporting similar research in adolescent swimmers [11,15]. This could have important practical implications considering that insufficient vitamin D is associated with impairments in muscle function, recovery, and immunity [1], which is an area for further research in swimming populations. Based on these results, highly trained adolescent swimmers in the UK should consider following standardised vitamin $D_3$ supplement protocols from October until March [30]; however, given the variable doses used in this study, the exact dose remains unclear.

While the use of vitamin $D_3$ supplements were mostly found to preserve 25(OH)D concentrations during the autumn and winter months, variable effects were observed with some doses. For example, supplementing with 2500 IU·day$^{-1}$ was thought to be an appropriate dose for adolescent swimmers [11], but upon consuming this amount, serum 25(OH)D concentrations either increased ($n = 2$), maintained ($n = 2$), or declined ($n = 1$). Such variable responses to this dose may have occurred for numerous reasons, including some swimmers (a) altering their dietary vitamin D and calcium intakes [31]; (b) changing their habitual UVB exposure (e.g., tanning beds) [32]; and/or (c) not adhering to their reported supplement intake [12]. However, these potential explanations are all speculative given that these confounding factors were not monitored. Nonetheless, these results support the findings of Dubnov-Raz et al. [11], who found that a similar 2000 IU·day$^{-1}$ strategy was only effective in 48% of adolescent swimmers. Alternatively, vitamin $D_3$ doses $\geq$4000 IU·day$^{-1}$ are thought to maintain a 'sufficient' 25(OH)D more consistently than doses of 1000–2000 IU·day$^{-1}$ [33,34], suggesting that higher doses (4000–5000 IU·day$^{-1}$) may be required when setting standardised supplement protocols. Such doses are well below the 'no observed adverse effect level' of 10,000 IU·day$^{-1}$ and are considered safe for children and adolescents [35,36].

Due to the variable vitamin $D_3$ supplement intakes being observed, the importance of nutrition education is also highlighted. All swimmers in this study received a ~20 min classroom-based education session regarding the roles of vitamin D and the challenges of maintaining vitamin D status in the winter, including a specific recommendation to supplement with 2000–5000 IU·day$^{-1}$ vitamin $D_3$ from October until March [8,9,11]. In addition, all care givers received educational material electronically, including supplement advice. However, this education method only resulted in 35% ($n = 7$) of swimmers reporting the use of vitamin $D_3$ supplements within the recommended range. Interestingly, this low adherence to vitamin $D_3$ supplement recommendations is commonplace, with many athletes either not perceiving themselves at risk of deficiency [13], lacking appropriate supplement knowledge to confidently buy the correct supplements [14], and/or not valuing the cost of vitamin $D_3$ supplements as a worthwhile investment [37], all of which point towards a flaw in the current methods used to transfer nutrition knowledge to practice in athletes. Indeed, previous work in this cohort identified that 20–30 min classroom-based education sessions increased sport nutrition knowledge [38], though whether this knowledge translates into meaningful dietary changes remains less clear [39]. Alternatively, future education strategies might have greater success at improving practical nutrition behaviours by specifically aiming education towards care givers, coaches, and support networks alongside the athlete [40], especially since the adolescent swimmers in this study were likely to have been reliant on care givers to purchase and administer vitamin $D_3$ supplements.

A limitation of this study was the use of dried blood spot cards, which were selected based on logistical and ethical considerations with this study cohort [41]. Previous research into vitamin D status in swimmers has analysed venous blood samples, given that 25(OH)D is largely found in the plasma [8,9,11,15]. However, this method requires specialist equipment and the expertise to perform venipuncture on adolescents, followed by the timely transportation of blood to a processing laboratory, which was not possible in this study [19]. The alternate use of blood spot cards meant that 25(OH)D was analysed from whole-capillary blood collected from the fingertip, which had to be corrected to account for sex-specific haematocrit levels [20]. This process often produces slightly lower 25(OH)D

concentrations than found in plasma (~1.7–8.0 nmol·L$^{-1}$), although the agreement between both measures is generally good [20,21,42]. Therefore, while this study's classification of swimmers' vitamin D status should be interpreted cautiously, the observed changes in serum 25(OH)D are thought to be reliable.

## 5. Conclusions

Overall, this study showed that highly trained adolescent swimmers in the UK are at risk of insufficient circulating vitamin D during winter months, which could negatively affect health and performance at this time. A secondary finding was that swimmers did not adhere to supplement recommendations, with only 50% of the cohort reporting the use of vitamin D$_3$ across the observation period (which were also highly variable in the doses used). Nonetheless, this supplementing sub-group better preserved their vitamin D status versus non-supplement users, supporting the use of standardised supplementation strategies for all swimmers across the autumn and winter training periods. Further research may, therefore, be required to develop contemporary vitamin D education strategies to improve adherence to supplement recommendations.

**Author Contributions:** Conceptualization, J.W.N. and L.A.G.; methodology, J.W.N. and M.A.B.; formal analysis, J.W.N.; investigation, J.W.N.; writing—original draft preparation, J.W.N.; writing—review and editing, M.A.B., M.C., A.L.K. and L.A.G.; visualization, J.W.N. All authors have read and agreed to the published version of the manuscript.

**Funding:** This research received no external funding.

**Institutional Review Board Statement:** This study was granted ethical approval by Birmingham City University (Newbury/7594/R(B)/2020/Aug/HELS FAEC) in accordance with the Declaration of Helsinki.

**Informed Consent Statement:** Informed consent was obtained from all participants involved in the study, including from their parents/guardians for those who were aged under 18 years.

**Data Availability Statement:** Data sets can be obtained through contacting the corresponding author.

**Acknowledgments:** We would like to acknowledge the support of Carl Grosvenor and Chris Littler from City of Birmingham Swimming Club for facilitating the research process.

**Conflicts of Interest:** The authors declare no conflict of interest.

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
