# Peer review of "An Observation of the Vitamin D Status in Highly Trained Adolescent Swimmers during the UK Autumn and Winter Months"

_physiologia, doi:10.3390/physiologia3030031_

Round 1
Reviewer 1 Report
Congratulations to the authors for the work developed so far. The issue of vitamin D deficiency is always up to date with a significant direct effect on sports performance.
Please consider the following suggestions to improve the quality of your work further.
Introduction: Please include a research hypothesis.
Discussion: According to previous studies, Vitamin D deficiency is higher for athletes who specialized in power sports. With this in mind did the authors consider the specialization of the participants i.e., 50 / 100m or longer swimming events? Also, female athletes are at greater risk of vitamin D deficiency. Please, add some lines of text to discuss this.
Author Response
Reviewer One
Congratulations to the authors for the work developed so far. The issue of vitamin D deficiency is always up to date with a significant direct effect on sports performance. Please consider the following suggestions to improve the quality of your work further.
JN Response – Thank you for your kind words and we appreciate your time taken to review and improve our manuscript.
Introduction: Please include a research hypothesis.
JN Response – A research hypothesis has now been added in lines x–x.
Discussion: According to previous studies, Vitamin D deficiency is higher for athletes who specialized in power sports. With this in mind did the authors consider the specialization of the participants i.e., 50 / 100m or longer swimming events?
JN Response – A specialisation comparison was not considered for this study since adolescent swimmers typically do not specialise in one event/distance. Furthermore, we are unsure whether this comparison would apply to the elite swimming context given that sprint swimmers will complete a similar duration and quantity of indoor training as their endurance counterparts (Pollock et al., 2019). Though, it does raise an interesting concept for future work.
Also, female athletes are at greater risk of vitamin D deficiency. Please, add some lines of text to discuss this.
JN Response – We are not sure there is sufficient evidence for us to make this claim (Fields et al., 2019; Geiker et al., 2017; Harju et al., 2022; Krzywański et al., 2020; Millward et al., 2020; Williams et al., 2020; Zürcher et al., 2018). Furthermore, our data appeared to show no difference in 25(OH)D between sexes in the autumn (male = 84.7 ± 22.9 vs. female = 86.5 ± 28.0 nmol·L-1), with the winter values being highly variable because of differences in supplementation (male = 67.5 ± 24.9 vs. female: 79.8 ± 42.8 nmol·L-1), as per the group the mean data.
Reviewer 2 Report
Introduction
Your introduction is well-structured. However, although you present the beneficial effects of Vitamin D in a general way and discuss its concentration in the human body, the question that arises is: what are the consequences of a deficiency in Vitamin D3? For instance, did this deficiency have any negative effects on the performance of swimmers or any other aspects?
120-124. Could you please explain how you collected the information? Furthermore, did you provide the participants with slides containing their answers that highlighted the significance of Vitamin D? If so, I am concerned that this action might introduce bias into your study, as participants could have developed a positive attitude towards Vitamin D based on the information presented. Consequently, this could potentially lead to an effect influenced by the placebo effect."
138. Mauchly’s test
144. it’s a = 0.05 not p
It would be helpful to present a diagram illustrating the study’s concept.
How did you split the groups?
Discussion
How did you explain the deficiency in serum 25(OH)D concentration independently of their Vitamin D consumption?
204. Benefited for what? Better performance, recovery after swimming training, etc.?
211. You should control this dropout in your study. Also, did you record their daily food or supplement consumption?
Author Response
Reviewer Two
Your introduction is well-structured. However, although you present the beneficial effects of Vitamin D in a general way and discuss its concentration in the human body, the question that arises is: what are the consequences of a deficiency in Vitamin D3? For instance, did this deficiency have any negative effects on the performance of swimmers or any other aspects?
JN Response – Thank you for complimenting our introduction to this manuscript, we appreciate the time you have given to review and strengthen our work. The purpose of our study was to observe the seasonal change in vitamin D status of adolescent swimmers, with the detection of performance, physical, or health markers beyond our scope. Hence, this introduction focussed on the seasonal changes in blood values, though we see that this may downplay the importance of our study. We have now briefly added some context to why these changes might affect performance/health in lines 70–71.
120-124. Could you please explain how you collected the information?
JN Response – During the collection of the winter blood data, the lead researcher asked the swimmers to declare if they had been using supplements across the observation period. This sentence in lines 131–132 has been amended to make this clearer to the reader.
Furthermore, did you provide the participants with slides containing their answers that highlighted the significance of Vitamin D? If so, I am concerned that this action might introduce bias into your study, as participants could have developed a positive attitude towards Vitamin D based on the information presented. Consequently, this could potentially lead to an effect influenced by the placebo effect."
JN Response – The swimmers were not questioned on whether they perceived vitamin D to be important and therefore did not receive any slides detailing their own specific answers. In contrary, all swimmers and primary care givers were given education/slides containing information that suggested that vitamin D was important, including a specific recommendation to ingest 2000–5000 IU·day-1 vitamin D3. (lines 124–130). The purpose of this was to develop a positive attitude/bias towards vitamin D, as this was thought to be ethical given that decreases in vitamin D are associated with declines in health and performance. This action produced the secondary purpose of this paper: to observe whether swimmers actually utilise the education given and/or adhere to supplement recommendations (i.e., lines 13–14, and lines 85–91).
- Mauchly’s test
JN Response – This has now been amended (lines 143–144).
- it’s a= 0.05 not p
JN Response – We do not want to alter this statement as statistical significance is typically presented as p = 0.05, including in Physiologia’s recent publications.
It would be helpful to present a diagram illustrating the study’s concept. How did you split the groups?
JN Response – The groups were split on whether they declared the use of vitamin D supplements, as per lines x–x. To make this clearer, a flowchart has been added after the statistical analysis section (line 162).
Discussion: How did you explain the deficiency in serum 25(OH)D concentration independently of their Vitamin D consumption?
JN Response – The key justification for the study was that swimmers may experience deficient/insufficient serum 25(OH)D during the winter months due to a decline in UVB availability and their large indoor training volumes (lines 40–48). This would therefore explain why declines would have occurred.
- Benefited for what? Better performance, recovery after swimming training, etc.?
JN Response – This sentence has now been amended in lines 236–239 to state “swimmers in the UK will likely require vitamin D3 supplementation to sustain sufficient serum 25(OH)D concentrations throughout the autumn and winter months”. We refrain from making any suggestion that a ‘sufficient’ status was beneficial for performance given we did not include a performance variable.
- You should control this dropout in your study. Also, did you record their daily food or supplement consumption?
JN Response – There were no dropouts in this study. This sentence referred to the 50% of the study sample who did not use vitamin D supplements (NONE sub-group), compared to the 50% who declared using supplements (VITD sub-group). Hopefully, this has now been made clearer through the flowchart requested.
Reviewer 3 Report
The article entitled "An Observation of the Vitamin D Status in Highly Trained Adolescent Swimmers During the UK Autumn and Winter Months" represents an insightful examination of the vitamin D status of a particular group of athletes–trained swimmers. Although the article makes a valuable contribution to the field of sports science and nutrition, there are both strengths and areas for improvement that deserve attention.
The selection of highly trained adolescent swimmers as the study population is appropriate, as these individuals are often engaged in an intense training program. On the other hand, it would be very good to have a control group of non-athletes (or non-swimmers)as well. A larger and more diverse sample size could improve the generalizability of the results.
Focusing on the fall and winter months is particularly important because these seasons are characterized by lower sunlight exposure, which can significantly affect the body's ability to synthesize vitamin D. This would allow for a more comprehensive understanding of seasonal variations in vitamin D levels and their potential impact on athletic performance and health. Please keep this in mind in the limitations.
In conclusion, the study "An Observation of the Vitamin D Status in Highly Trained Adolescent Swimmers During the UK Autumn and Winter Months" investigates the relationship between vitamin D status and seasonal changes in a specific group of athletes.
Author Response
Reviewer Three
The article entitled "An Observation of the Vitamin D Status in Highly Trained Adolescent Swimmers During the UK Autumn and Winter Months" represents an insightful examination of the vitamin D status of a particular group of athletes–trained swimmers. Although the article makes a valuable contribution to the field of sports science and nutrition, there are both strengths and areas for improvement that deserve attention.
JN Response – We would like to thank the reviewer for taking the time to review our manuscript and offer their valuable contributions to strengthen our work.
The selection of highly trained adolescent swimmers as the study population is appropriate, as these individuals are often engaged in an intense training program. On the other hand, it would be very good to have a control group of non-athletes (or non-swimmers) as well. A larger and more diverse sample size could improve the generalizability of the results.
JN Response – While we agree that a larger and more diverse sample size would undoubtedly have strengthened our results, we were restricted by our available funding. We therefore decided to focus our resources on measuring the 25(OH)D concentrations solely on adolescent swimmers, which was more appropriate for answering our research question. We have now added a short section in lines 308–313 to explain our budget limitations and how this affected our methodology.
Focusing on the fall and winter months is particularly important because these seasons are characterized by lower sunlight exposure, which can significantly affect the body's ability to synthesize vitamin D. This would allow for a more comprehensive understanding of seasonal variations in vitamin D levels and their potential impact on athletic performance and health. Please keep this in mind in the limitations.
JN Response – The authors agree with this comment and due to our limited funding, this is why this manuscript focussed on gathering a larger participant sample at the fall and winter months as opposed to a smaller study sample across the entire season. In line with the previous comment, we have now added this limitation to lines 308–313.
In conclusion, the study "An Observation of the Vitamin D Status in Highly Trained Adolescent Swimmers During the UK Autumn and Winter Months" investigates the relationship between vitamin D status and seasonal changes in a specific group of athletes.
Reviewer 4 Report
The manuscript presented for review is an interesting research work on serum vitamin 25(OH)D concentrations in young UK swimmers. Variability related to the season was considered, as well as athletes' supplementation of this vitamin. The parts of the manuscript do not raise any major concerns for me. Please find my detailed comments below.
Abstract
It seems that a more important goal is to indicate changes in serum vitamin 25(OH)D concentrations over time than adherence to supplementation. Hence, I would suggest swapping the objectives (a) and (b) listed in the abstract
Methods
2.2: Please add information whether the blood draw was done by one of researchers or the swimmers and their care givers on their own.
Author Response
Reviewer Four
The manuscript presented for review is an interesting research work on serum vitamin 25(OH)D concentrations in young UK swimmers. Variability related to the season was considered, as well as athletes' supplementation of this vitamin. The parts of the manuscript do not raise any major concerns for me. Please find my detailed comments below.
JN Response – We are pleased to have presented a manuscript without many major concerns and would like to thank the reviewer for their time and effort in providing constructive feedback.
Abstract: It seems that a more important goal is to indicate changes in serum vitamin 25(OH)D concentrations over time than adherence to supplementation. Hence, I would suggest swapping the objectives (a) and (b) listed in the abstract
JN Response – The authors agree with this observation and have now made this change in lines 12–14.
Methods, 2.2: Please add information whether the blood draw was done by one of researchers or the swimmers and their care givers on their own.
JN Response – All blood draw was completed by lead researcher for consistency, which has now been made clear in lines 114–115.
Round 2
Reviewer 2 Report
149. In the majority of analyses, an alpha of 0.05 is used as the cutoff for significance. Then in your results you can use p.
Author Response
Line 157 - we thank the reviewer for the comment and we have now included a line to state that significance was accepted as p<0.05.